



# GCM clouds and actual clouds as seen from different space lidars : towards a long-term assessment of cloud representation in GCMs using lidar simulator

Marie-Laure Roussel[1], Hélène Chepfer[1], Zacharie Titus[1], and Marine Bonazzola[1]

[1]Laboratoire de Météorologie Dynamique, CNRS

**Correspondence:** Marie-Laure Roussel (mlroussel@lmd.ipsl.fr)

**Abstract.**

In Earth's radiative budget, clouds play a central role but their representation in General Circulation Models (GCMs) remains a major source of uncertainty for climate projection. Here, we used spaceborne lidar observations to assess cloud distribution in the IPSL-CM6-LR model using the CFMIP Observation Simulator Package (COSP). We focused on the lidars onboard

CALIPSO and AEOLUS satellites during 2006-2023 and 2018-2023. While CALIPSO has been widely used for GCMs evaluation, ALADIN was originally designed for wind profiling. However, studies have demonstrated its potential to retrieve reliable cloud profiles. A new module was developed to simulate AEOLUS observations within COSP-lidar, extending original implementations made for CALIPSO, including wavelength change (532nm to 355 nm), viewing geometry (35° off-nadir) and specific parameters adjustments related to sensivity and resolution. We compared our simulations to 1-year observations for

both instruments. " Results show that AEOLUS observations can effectively evaluate clouds in GCMs, as it shows similar cloud fraction biases in IPSL-CM6-LR to those obtained with CALIPSO. Significant underestimations of low (up to 20%) and high clouds in certain regions (e.g. warm pool) were re-assessed for this model. Sensitivity analyses highlighted the small role of instrument-specific parameters in COSP-lidar: viewing geometry, multiple scattering coefficient and cloud detection threshold (associated with wavelength and sensivity). This work lays the foundation for a consistent multi-decades evaluation

of cloud representation using different lidar missions, and supports the integration of EarthCARE/ATLID in COSP-lidar for further model evaluation.

## 1  Introduction

Clouds play a crucial role in Earth's energy budget by modulating the balance between incoming solar radiation and outgoing thermal radiation. However, cloud feedback processes remain one of the largest sources of uncertainty in climate projections

(Boucher et al., 2020). Understanding cloud properties and their interactions with radiation is essential for improving climate predictions, as even small changes in cloud characteristics can have significant impacts on global temperature and circulation in the atmosphere (Zelinka et al., 2020; Sherwood et al., 2020). To improve and refine climate models, observational data are indispensable, but measuring clouds precisely at the global scale and capturing their detailed vertical distribution present significant challenges. Spaceborne instruments such as LiDAR (Light Detection And Ranging, e.g. (Hunt et al., 2009; Wehr





et al., 2023)) provide a unique capability to retrieve cloud properties almost everywhere around the globe with high vertical resolution, offering precious insights into cloud properties influencing cloud radiative effects.

However, the validation of cloud representations within Global Climate Models (GCMs) through observational data presents inherent complexities, primarily coming from disparities in model cloud definition and spatial resolution, but also from space-borne instrument configurations. The CFMIP Observation Simulator Package (COSP) (Bodas-Salcedo et al., 2011; Swales

et al., 2018) is a tool facilitating direct model to observations comparison by simulating instrument-specific measurements as they would be acquired above the atmosphere modeled by a GCM. Extending the COSP-lidar algorithm from previous developments made for CALIOP, the LiDAR of CALIPSO (Chepfer et al., 2008; Guzman et al., 2017; Bonazzola et al., 2023) which was operating from 2008 to 2023, is a key point of this work. We have updated COSP to accurately simulate measurements from different instrumental characteristics than CALIPSO, and especially those of the 355 nm Doppler lidar (ALADIN)

onboard AEOLUS from 2018 to 2023.

By incorporating the capability to simulate measurements from an additional LiDAR in COSP, our goal is to enable comparative studies with models across a broader range of instruments, supporting the on-going evaluation of cloud description and parametrization in CMIP models and multi-model assessments (Cesana and Chepfer, 2013; Cesana et al., 2024; Konsta et al., 2022), and to build a continuous and realistic long-term time-serie of simulations of spaceborne LiDAR observations of

clouds from successive spaceborne lidars. Furthermore, these advancements will directly contribute to the final implementation of the EarthCARE lidar (ATLID, (Wehr et al., 2023; Donovan et al., 2024)) module in the COSP algorithm, given its shared instrumental characteristics with AEOLUS (355 nm wavelength, High Spectral Resolution capabilities) (Reverdy et al., 2015; Feofilov et al., 2023).

In this study, we used the outputs from the model of the "Laboratoire de Météorologie Dynamique" (LMD) named LMDZ

(for its zooming capability), that is involved in the CMIP (Climate Model Intercomparison Project) experiments as it is the atmospheric part (Hourdin et al., 2020) of the global model of the IPSL (Institut Pierre-Simon Laplace). In particular, the bias of this model has been extensively evaluated by comparing LMDZ+COSP-lidar/CALIPSO simulations to CALIPSO observations (Cesana et al., 2022). These studies have shown that cloud covers are underestimated in LMDZ with respect to CALIOP measurements, despite the significant improvement of parameterizations from version 5A to 6A (Madeleine et al.,

50 2020).

In this context, we conducted COSP-lidar simulations using the LMDZ atmospheric model to compare its cloud representation against observations from two spaceborne lidars: AEOLUS (2020) CALIPSO (2008). These comparisons aim to assess whether AEOLUS measurements - despite its initial goal of measuring winds and specific instrumental characteristics - can be used like CALIPSO to evaluate the quality of cloud simulations in climate models. Ultimately, our objective is to enable a

uniform, long-term, and realistic evaluation framework for cloud representation in models using different spaceborne lidars - CALIPSO, AEOLUS, and now including ATLID on board EarthCARE since 2024.

The paper is organized as follows: Section 2 presents the lidar-based cloud observations from CALIPSO and AEOLUS, and discusses the representativity of the selected years. Section 3 describes the cloud simulations using COSP-lidar, the model used for the inputs (LMDZ), and the analysis of the parameters we modified to simulate AEOLUS-like observations. Section





4 presents the results, including comparisons between AEOLUS and CALIPSO observations, simulations from COSP-lidar driven by LMDZ in CALIPSO and AEOLUS configurations, and their differences with respect to observations. We conclude in Section 5 with a synthesis of the main findings and future perspectives for extending this work toward EarthCARE applications.

## 2 Cloud observations from space lidars

### 2.1 Measurements from CALIPSO space lidar

The CALIOP (Cloud-Aerosol Lidar with Orthogonal Polarization) instrument on board CALIPSO (Cloud-Aerosol Lidar and Infrared Pathfinder Satellite Observations) satellite is a spaceborne polarization-sensitive lidar operating between 2006 and 2023 designed to provide vertical profiles of clouds and aerosols in the Earth's atmosphere. With its 532 nm and 1064 nm channels, CALIPSO measures backscattered laser signals to derive cloud and aerosol properties, such as optical depth and layer height. CALIPSO operates in a sun-synchronous orbit and crosses the equator at approximately 13:30 local solar time 70 on its ascending node and at 01:30 on its descending node. It operates with a near-nadir viewing geometry, with an inclinaison of 3° since 2008. Between 2011 and 2020, CALIPSO emitted low energy laser shots resulting in a reduced signal strength degrading the quality of the measurement dataset (Hunt et al., 2009). This issue was particularly associated with the satellite's passage through the South Atlantic Anomaly (SAA) which is a region over which the onboard electronics can be disrupted by the exposure to high-energy particles (Noel et al., 2014). Here we use the GOCCP v3.14 (GCM-Oriented CALIPSO Cloud 75 Product) dataset, derived from CALIPSO observations (Chepfer et al., 2010; Cesana and Chepfer, 2013). The low energy events have been identified and properly accounted for in this version, ensuring reliable and homogenous quality of the dataset (Chepfer et al., 2025). The cloud detection is done at the higher resolution of the instrument, 75 m cross track and 330 m along track, but here we use the low, mid and high levels cloud covers and cloud fractions profiles at a horizontal resolution of 2° × 2° and vertical resolution of 480 meters. Global coverage is provided up to 82° latitude and we choose to use monthly datasets 80 of daily averages, facilitating the evaluation of cloud representation between the observations and the outputs from the climate model.

In this study, CALIPSO observations from the year 2008 are primarily used to ensure consistency with the model simulations (see Section 3.1). A direct comparison is thus made between CALIPSO measurements and model outputs for 2008, providing a coherent framework for evaluation. To assess whether such a comparison can be extended to other years, a representativeness 85 analysis is conducted using CALIPSO data from the 2008–2018 decade, as well as from 2020, a year during which CALIPSO was still in operation (see Section 2.3). This analysis aims to determine whether the cloud measurements from AEOLUS in 2020 can be meaningfully compared to simulations from 2008, under the assumption that atmospheric conditions remain sufficiently similar.

The years 2007 and 2016 are excluded from the analysis due to data quality limitations. Specifically in 2007, CALIPSO 90 had not yet adopted its final 3° inclination configuration, while the 2016 year is incomplete in the dataset due to a lack of measurements in February, compromising the reliability of temporal averaging for that year.





## 2.2 Measurements from AEOLUS space lidar

The Atmospheric Laser Doppler Instrument (ALADIN) is a 355 nm High Spectral Resolution (HSRL) spaceborne Doppler Wind LiDAR on board the AEOLUS satellite, primarily designed to retrieve horizontal wind profiles. It also operates in a sun-synchronous orbit and crosses the equator at approximately 18:00 local solar time on its ascending node and at 06:00 on its descending node. It also provides valuable cloud profile observations (FLAMANT et al., 2008) at a nominal horizontal resolution of 87 km. While it is insufficient for detecting smaller cloud structures, such as shallow cumulus (Feofilov et al., 2022), recent studies have demonstrated that cloud detection is feasible at the full horizontal resolution of 3 km along the orbit track (Wang et al., 2024) of the instrument.

Building on these advances, a similar cloud method has been pursued by Titus et al. (2025) with key modifications compared to Wang et al. (2024). For example, it compensates for the absence of a cross-polar component using a climatology derived from CALIPSO-GOCCP observations and systematically discards hot pixels to enhance detection reliability. For more information about the method, please refer to Titus et al. (2025). Cloud and wind profiles are distributed across different AEOLUS data products at varying spatial resolutions, these datasets have been merged to ensure a fully integrated and usable dataset enabling cloud-wind interactions studies (Titus, 2024).

AEOLUS operates with a laser pointed 35° off-nadir and perpendicular to the satellite track, away from the sun. For consistency with CALIPSO-GOCCP (Chepfer et al., 2010), the reprocessed dataset features a vertical resolution of 480 m and a horizontal resolution of 3 km along the orbit track—the latter matching the highest possible resolution for cloud detection with AEOLUS. The method retrieves cloud fraction profiles at 3 km resolution using AEOLUS Level 1A observations, contributing to improved characterization of cloud structures in spaceborne lidar measurements.

The satellite was launched in 2018 and the mission ended in 2023. In this study, we used the dataset provided by Titus et al. 2025 for the full year 2020.

## 2.3 Representativity of the selected year

Here we compare observational data of the 2020 year from AEOLUS with CALIPSO ones of the year 2008. In parallel, these measurements are also used to evaluate clouds simulated by the LMDZ model. As this approach involves comparing datasets from different years, it is essential to assess whether interannual variability in cloud distribution introduces a significant bias. To address this issue, we estimate the interannual variability of cloud fraction and cloud cover at high,mid and low levels using the CALIPSO observational time series from 2008 to 2018, which provides a consistent and long-term reference for cloud distribution.

Figure 1 shows the median (left column) and the standard deviation (right column) of the yearly means (between 2008 and 2018) of cloud covers respectively at high (mean from 8 to 18 km), mid (mean from 4 to 8 km) and low (mean from 0 to 4 km) altitudes. The global median of high cloud cover is 31%, with a small interannual variability of 4%, but we noticed both very large regional values of high cloud cover - as in the warm pool region where it goes up to 52% - and very little values - as in stratocumulus-dominated areas where it does not exceed 20%. For mid-level cloud cover, the global median is around 21%,





| | Global mean of yearly means (spatial min/max) | | DJF | MAM | JJA | SON |
|---|---|---|---|---|---|---|
| | Median over 10 years | Standard deviation | Median over 10 years | Median over 10 years | Median over 10 years | Median over 10 years |
| High cloud cover | 31% (6,2-82) | 4,3% (0,9-13) | 31% | 31% | 31% | 30% |
| Mid cloud cover | 21% (0,7-57) | 3,0% (0,2-7,6) | 21% | 20% | 20% | 20% |
| Low cloud cover | 40% (0-85) | 3,5% (0-24) | 39% | 39% | 39% | 40% |

**Table 1.** Global mean of yearly and seasonal means of high/mid/low cloud covers for CALIPSO-GOCCP over the 10 years (DJF = December, January, February ; MAM = March, April, May ; JJA = June, July, August ; SON = September, October, November

accompanied by a lower interannual variability of approximately 3%. In contrast, low cloud cover exhibits a global median of 40%, with pronounced spatial variability—ranging from an average of 52% in stratocumulus regions to about 19% over the Indian Ocean.

The mean cloud cover for the years 2008 and 2020 are presented in the figure 3 along with the geographic regions where values deviate significantly from the CALIPSO decadal mean. Areas exceeding three standard deviations from the 10-year

average are shaded in gray, indicating that the values are not statistically significant (at the 99.7% confidence level). Based on the assumption of normally distributed, independent, and comparable data, the values respectively observed in 2008 and 2020 that fall within ±3 standard deviations from the 10-year mean indicate no statistically significant deviation from the 2008-2018 decade. The mean cloud cover values for the years 2008 and 2020 at high, middle, and low atmospheric levels are 30%, 20%, and 39% for 2008, and 33%, 22%, and 40% for 2020, respectively. All these values fall within the confidence range defined

by the median and standard deviation provided in Table 1, thereby allowing for a reliable year-to-year comparison between 2008 and 2020. Also, table 1 shows that there is no significant change at the global scale of the median when studying seasonal means of cloud covers.

Figure 2 displays the zonal annual mean of the median cloud fraction profile over the 2008–2018 decade and the corresponding standard deviation for the same period. Interannual variability is most pronounced in high-altitude clouds near the equator

and in low-level clouds (below 3 km altitude) across all latitudes. These regions generally correspond to areas with higher cloud fractions, where global values always exceed 15% and can locally surpass 35%. Then, figure 4 presents the zonal profiles of annual mean cloud fraction for the year 2008 (a), for 2020 (b), and the difference between the two (c). As in Figure 2, the shaded areas indicate latitudes and altitudes where the values for these specific years are not representative of the 2008–2018 decadal average. The Arctic low levels cloud fractions appear to be quite different between 2008 and 2020, but except for

the region between 25°S and 5°S (and above 14000 meters altitude) - affected throughout the atmospheric column by the instrumental issue previously mentioned - all cloud fraction values for 2008 and 2020 are considered statistically significant.



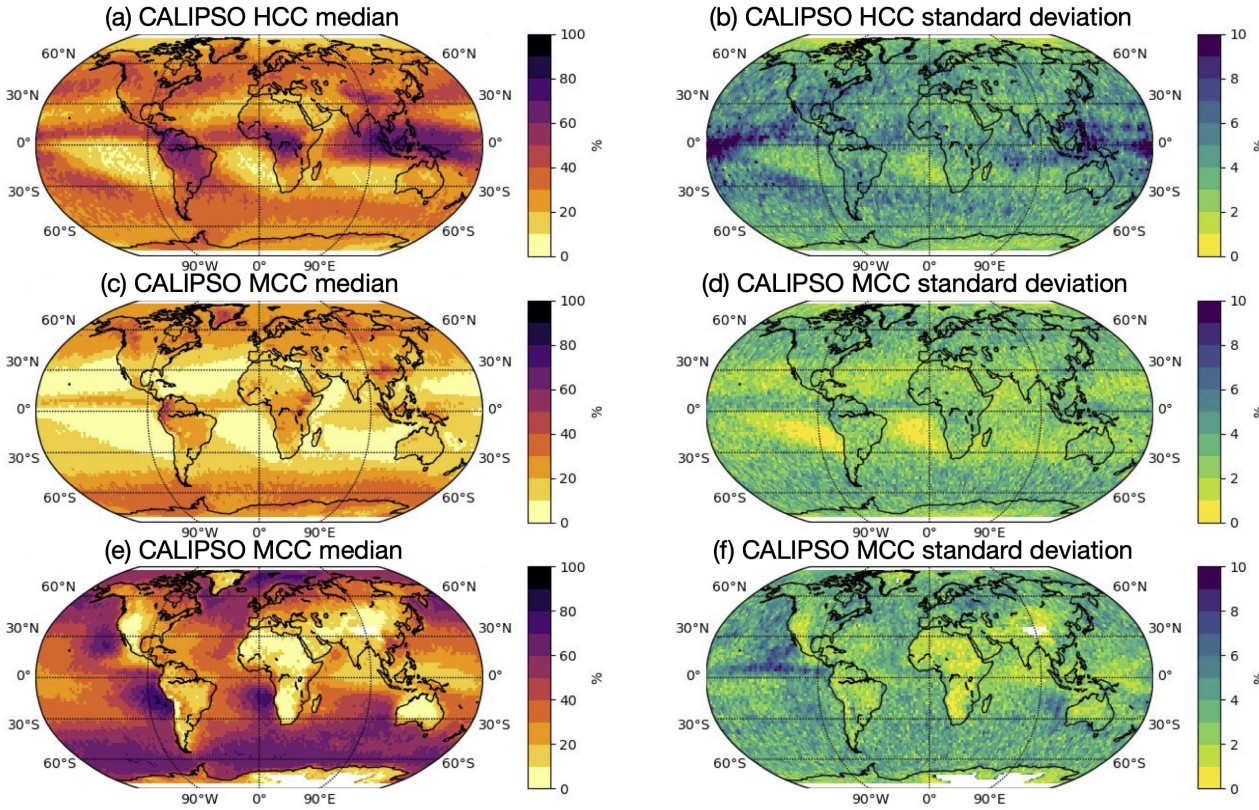

**Figure 1.** High (a,b) mid (c,d) and low (e,f) yearly means of cloud covers medians (left column) and standard deviations (right column) by CALIPSO between 2008 and 2018. HCC = High Cloud Cover (8-18 km), MCC = Mid Cloud Cover (4-8 km), LCC = Low Cloud Cover (0-4 km).

## 3 Model clouds seen from 2 different space lidars

### 3.1 The COSP-LIDAR algorithm: basics and references

The CFMIP Observation Simulator Package (COSP) is a tool designed to generate synthetic observations from remote sensing instruments by using model output variables as inputs. This approach avoids discrepancies in variable definitions and spatial resolution that typically arise when comparing model outputs with instrument measurements. Bodas-Salcedo et al. 2011 details the main steps involved in the algorithm. COSP can also be implemented directly within GCMs, as described in Swales et al. (2018). The interface of COSP is modular and adaptable to a wide range of satellite or in-situ instruments, and it has evolved with successive developments across versions 1 and 2 enabling broader applications.

The lidar simulation component (COSP-lidar) was initially developed to replicate measurements from CALIOP, the lidar onboard the CALIPSO satellite (Chiriaco et al., 2006; Chepfer et al., 2007). Over time, it has been enhanced to better mimic





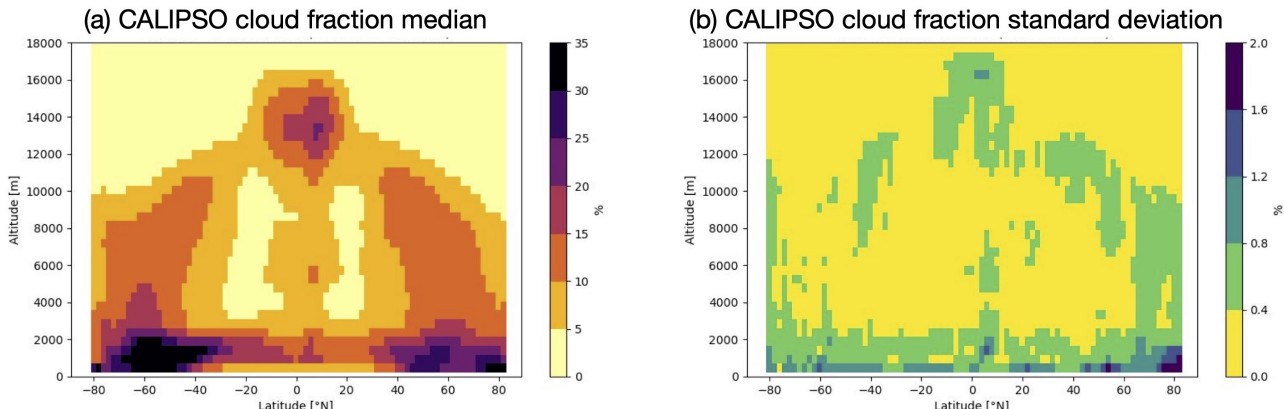

**Figure 2.** Zonal mean of yearly mean of (a) median and (b) standard deviation cloud fraction by CALIPSO between 2008 and 2018

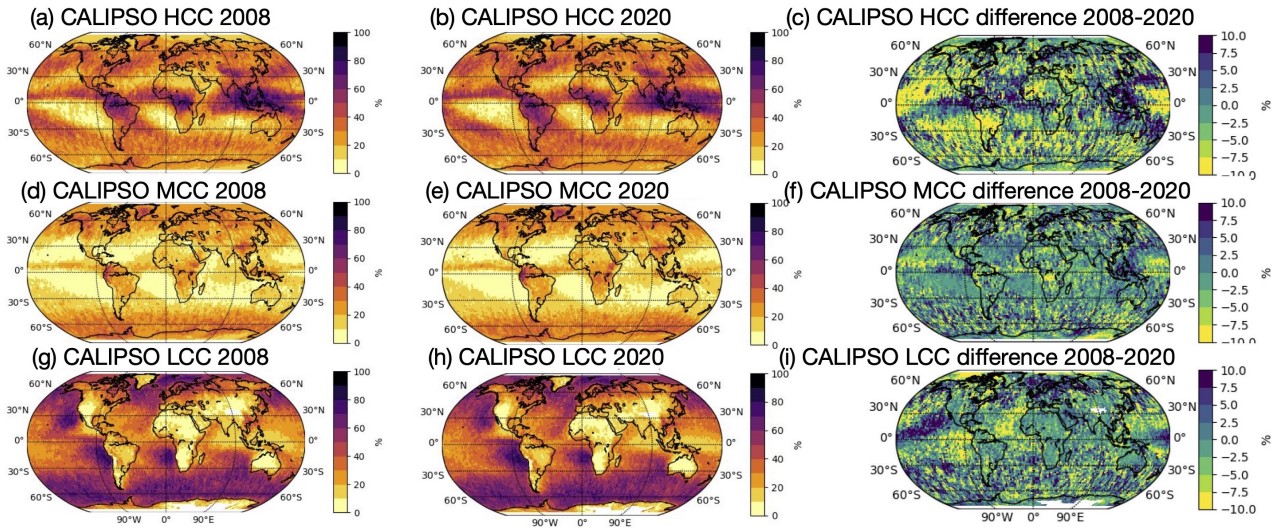

**Figure 3.** High (a,b,c) mid (d,e,f) low (g,h,i) cloud covers for 2008 (left column), 2020 (middle column) and 2008-2020 difference (right column) by CALIPSO. Grey zones indicate not significant areas (where cloud cover difference is higher than the 2008-2020 mean + 3 standard deviation). HCC = High Cloud Cover (8-18 km), MCC = Mid Cloud Cover (4-8 km), LCC = Low Cloud Cover (0-4 km).

specific instrument capabilities, including cloud fraction and 3D cloud structure (Chepfer et al., 2008), cloud phase differentiation (Cesana and Chepfer, 2013), opaque cloud detection (Guzman et al., 2017), and aerosol characterization (Bonazzola et al., 2023). For spaceborne lidar applications, COSP provides a crucial framework for scale-aware and definition-consistent com-

parisons between modeled and observed cloud properties, particularly valuable for building long-term multi-lidar simulation datasets. It has already been extensively used to evaluate cloud representation in various models, including those participat-





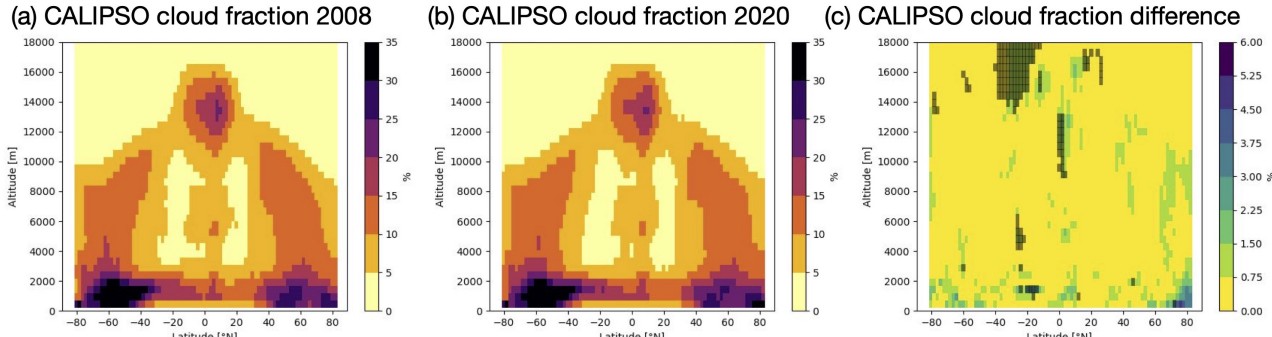

**Figure 4.** Zonal mean of yearly means of cloud fraction in (a) 2008 (b) 2020, and the (c) 2008-2020 difference, by CALIPSO. Grey zones indicate not significant areas (where cloud fraction difference is higher than the 2008-2020 mean + 3 standard deviation)

ing in CMIP5 and CMIP6, facilitating consistent multi-model analyses and comparisons with observations (Nam et al., 2014; Williams and Bodas-Salcedo, 2017; Morrison et al., 2019; Kay et al., 2012; Konsta et al., 2022; Cesana et al., 2024).

## 3.2 Model outputs from LMDZ / IPSL-CM6-LR

In this study, we employed the offline version of COSP, which involves driving the simulator using output variables from a climate model. We selected the model developed by IPSL at the lowest resolution (143 x 144 x 79 grid corresponding to a 200 km horizontal resolution) for the whole 2008 year in the CMIP6 amip experiment (IPSL-CM6-LR, (Eyring et al., 2016)) made available on the database of the spirit cluster (see https://esgf-node.ipsl.upmc.fr/search/cmip6-ipsl/), at CF3hr or CFday frequency (depending on the availability of the variables - see next table for details). All the variables at CF3hr frequency 170 have been averaged to get a uniform daily dataset as input for COSP simulations. Table A2 in appendix lists all the variables required to run the lidar simulator, along with their descriptions, dimensions, and the corresponding variable names in the different environments. Some variables that have been put to zero in input of COSP because they are not necessary in the lidar simulation (and not present in the CMIP6 database for the IPSL-CM6-LR model): *mr_ozone*, *dtau_s*, *dtau_c*, *dem_s*, *dem_c*.

## 3.3 New developments in COSP-LIDAR for AEOLUS

Building on previous developments that enabled optical computations in the cloud module at the 355 nm wavelength for preparing the lidar on board the EarthCARE satellite, ATLID, (Reverdy et al., 2015; Feofilov et al., 2023) we have extended the algorithm to support the ALADIN instrument onboard AEOLUS. Since both ATLID and ALADIN operate at the same wavelength, we can reuse most of the parameter definitions, especially optical parameters, previously established for 355 nm. However, it was necessary to adjust the cloud detection threshold S, which differs between instruments due to their respective 180 sensitivities and horizontal resolutions, as well as the multiple scattering coefficient n. In the case of AEOLUS, the instrument's Line of Sight (LOS) - with an inclinaison of i = 35° - also must be taken into account in the cloud-related calculations because the laser beam travels through a longer atmospheric path compared to a vertical observation. To accurately simulate





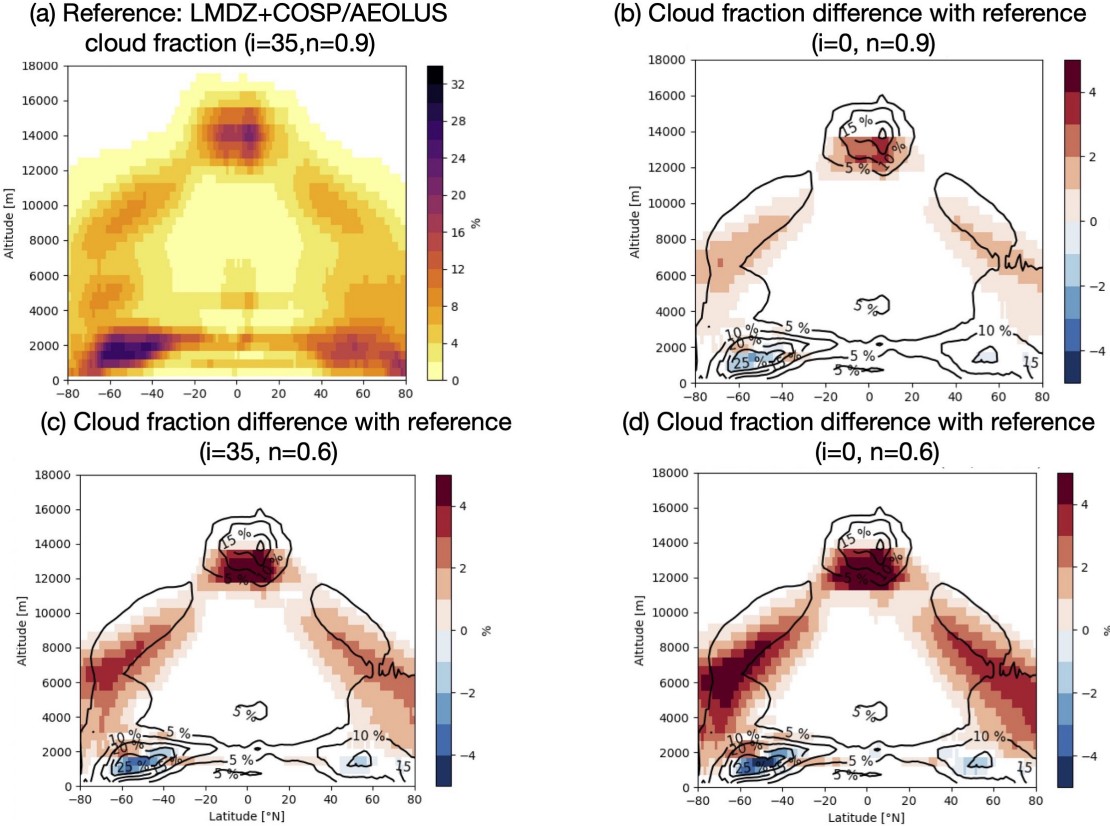

**Figure 5.** Influence of the inclinaison (i) and multiple scattering coefficient (n) on the cloud fraction. Cloud fraction lower than 0.1 (for the reference) and absolute values of cloud fraction difference lower than 0.5 are masked.

the instrument's measurements in the COSP algorithm, we have to replicate the instrument geometry. Therefore, variables 1 to 4 in the table A2 in appendix are multiplied by $1/\cos(35°)$ to adjust for the longer optical path caused by the inclination.

The following section (3.3) examines the influences of these parameters (wavelength, inclinaison, cloud detection threshold, multiple scattering coefficient) on the outputs generated by the lidar simulation. More information on the developments are given in Appendix in table A1.

## 3.4 Sensitivity tests in COSP-LIDAR

In this section we address the topic of the influence of adjustable parameters related to cloud computation in COSP-lidar.

We conducted simulations using input data (for the full 2008 year) from the IPSL-CM6-LR model and modifying for each simulation one of the following parameters in the COSP algorithm: the laser inclinaison **i**, the multiple scattering coefficient **n**, the cloud detection threshold **s**.





### 3.4.1 Laser inclinaison

Figures 5 (b) and (d) illustrate the impact of the inclination angle on the annually averaged cloud fraction simulated for a
lidar operating at 355 nm, with a fixed multiple scattering coefficient of n=0.9 and a fixed detection threshold of s=1.84.
Red (respectively blue) indicates a positive (respectively negative) difference in cloud fraction with respect to the simulation
of reference (chosen with i=35). As AEOLUS measurements are performed at an off-nadir angle, the laser signal travels
a longer optical path through the atmosphere. This increased path length leads to greater signal attenuation, resulting in a
lower attenuated backscatter (ATB) thus to a lower cloud fraction measured compared to CALIPSO without inclinaison of the
instrument. This is consistent with the slightly higher (between 0 and 3%) cloud fraction simulated at all altitudes when the
35° inclinaison is removed, except for low level clouds around 2000 meters altitude. We particularly want to highlight the fact
that the inclination does not alter the global spatial distribution of clouds.

### 3.4.2 Multiple scattering coefficient

The difference of the values of the multiple scattering coefficient between two instruments is primarily due to their various
instrumental characteristics, such as footprint size and receiver field of view. n=1 corresponds to single scattering and this
value is decreasing as the effect of multiple scattering increases. For CALIPSO, the value of n has been previously investi-
gated and set to n=0.7 in COSP-lidar/CALIPSO (Garnier et al., 2015). In the literature, the selected values respectively for
EarthCARE/ATLID and AEOLUS are n=0.6 and n=0.9 (Reverdy et al., 2015; Feofilov et al., 2024).

Figure 5 (c) shows that the influence of the multiple scattering coefficient modification, with a fixed inclinaison of i=35° and
a fixed cloud detection threshold of s=1.84 (see Section 3.3.c), is bigger than the one of the inclinaison. The simulated cloud
fraction also decreases at all altitudes when the multiple scattering coefficient increases from n=0.6 to n=0.9, except in the
lower atmospheric layers below 2000 meters altitude, where cloud fraction values exceed 25%. Above 3000 meters altitude,
the reduction in cloud fraction ranges from approximately 2% to 5%. Figure 5 (d) confirms that the impact of simultaneously
varying both parameters is consistent with the cumulative effect of their individual contributions observed previously. This last
configuration (i = 0, n = 0.6, l = 355 nm) is representative of a simulation closely matching the measurement conditions of the
ATLID instrument. Finally, Table 2 details the global annual means of high, mid, and low-level cloud covers for the various
tested configurations, in which the inclinaison and the multiple scattering coefficient are added independently, while keeping the
detection threshold and wavelength fixed (s = 1.84 and l = 355 nm). The last column, corresponding to a simulated configuration
closely matching CALIPSO observational conditions, is included for reference. The results indicate that introducing both the
inclinaison and the increase in the multiple scattering coefficient lead to a decrease in cloud cover values by approximately 1
to 3%.

### 3.4.3 Detection threshold (horizontal resolution)

In the COSP-lidar algorithm, a fixed value s0 of scattering ratio (SR) is used as a threshold to identify cloudy layers (layers
with SR>s0 are flagged cloudy). Simulations are performed using a reference cloud detection threshold value of s0=1.84 in





| Global mean | i=35, eta=0.9 l=355 AEOLUS | i=35, eta=0.6 l=355 | i=0, eta=0.6 l=355 EarthCARE/ATLID | i=0, eta=0.9 l=355 | i=0, eta=0.7 l=532 CALIPSO |
|---|---|---|---|---|---|
| High cloud cover | **13%** | 14% | **14%** | 14% | **14%** |
| Mid cloud cover | **15%** | 17% | **18%** | 16% | **18%** |
| Low cloud cover | **39%** | 40% | **40%** | 39% | **40%** |

**Table 2.** Global mean of high/mid/low cloud covers with various configurations of inclinaison, multiple scattering coefficient and wavelength (with fixed s=1.84 for the 355 nm simulation and s=5.0 for the 532 nm simulation)

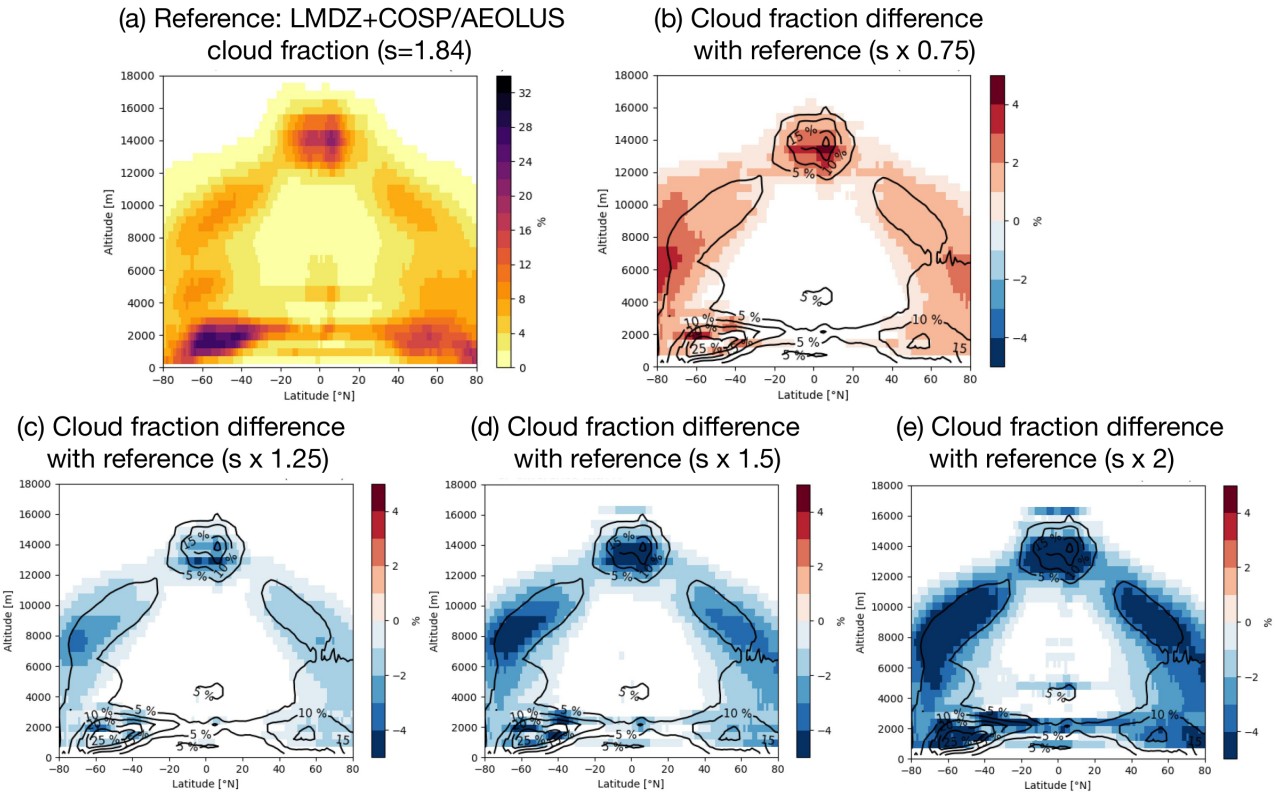

**Figure 6.** Influence of the cloud detection threshold (s). Cloud fraction lower than 0.1 and absolute values of cloud fraction difference lower than 0.5 are masked.

COSP-lidar/AEOLUS, as well as scaled values of s0 (0.75 × s0, 1.25 × s0 , 1.5 × s0 and 2 × s0). We set the multiple scattering coefficient to n=0.9 and the inclinaison of i=35° to replicate the measurements conditions of AEOLUS in all the simulations presented in this paragraph. The choice of s0=1.84 for AEOLUS is motivated by its operating conditions, measuring during the transition between day and night. Additionally, its 355 nm wavelength leads to an attenuated molecular backscatter signal





| Global mean | s0 x 0.75 | **s=1.84** | s0 x 1.25 | s0 x 1.5 | s0 x 2 |
|---|---|---|---|---|---|
| High cloud cover | 15% (+2) | **13%** | 12% (-1) | 10% (-3) | 7.4% (-5.6) |
| Mid cloud cover | 17% (+2) | **15%** | 14% (-1) | 12% (-3) | 11% (-4) |
| Low cloud cover | 41% (+2) | **39%** | 38% (-1) | 37% (-2) | 33% (-6) |

**Table 3.** Global mean of high/mid/low cloud covers with various configurations of cloud detection threshold

that is approximately five times lower than that observed at 532 nm (as by CALIPSO). This is due to the fact that the molecular
attenuated backscatter ($ATB\_mol$) is inversely proportional to the fourth power of the wavelength, in accordance with Rayleigh
scattering theory. As a result, the scattering ratio ($SR = ATB / ATB\_mol$) at 355 nm is about five times smaller than at 532
nm, given that the total attenuated backscatter ($ATB$) from cloud particles is insensitive to the wavelength as the particle sizes
involved are significantly larger than 355 and 532 nm. Consequently, the appropriate threshold for cloud detection at 355 nm is
around 5 times lower than the one at 532 nm. This threshold value is supported by the analysis of (Reverdy et al., 2015), who
estimated the cloud detection threshold for ATLID—operating at the same wavelength under nighttime (s=1.84) conditions.

Figure 6 presents the differences in cloud fraction obtained from various simulations as the cloud detection threshold varies,
with respect to the reference threshold s0=1.84. Simulations using bigger thresholds than s0 reveal that increasing this pa-
rameter leads to a systematic reduction in cloud fraction, both horizontally across the globe and vertically throughout the
atmospheric column. This decrease is particularly pronounced in regions characterized by high cloud fractions, in the lower
troposphere (around 2000 meters altitude) at latitudes between 60° and 40°, and at higher altitudes (around 14000 meters
altitude) near the equator. Only the simulation using a detection threshold lower than s0 (see figure 6 (b)) exhibits an over-
all increase in cloud fraction. The analysis of cloud cover at high, mid, and low altitudes confirms the result: increasing the
detection threshold leads to a reduction in cloud cover across all altitude levels uniformly. This result is expected as a higher
cloud detection threshold implies that less small attenuated backscatter signals are identified, leading to a lower measured
cloud fraction and potentially to undetected clouds. It is also observed that further increasing the threshold has a limited impact
on the results, whereas even a slight decrease in the threshold induces a more significant effect. For instance, multiplying the
cloud detection threshold by 0.75 produces a change in cloud cover of similar magnitude to that resulting from multiplying by
a factor of 1.5, meaning that LMDZ produces a lot of optically thin clouds.

In conclusion to these sensibility tests, we choose for the COSP-lidar/AEOLUS algorithm to keep the parameters n=0.9
and s=1.84 as prescribed in the literature (Reverdy et al., 2015; Feofilov et al., 2022) The results presented in the following
part of this article are based on the simulation performed using this configuration, which incorporates these parameters with a
wavelength of l = 355 nm and a i=35° instrument inclinaison, to accurately represent the measurements conditions of AEOLUS.
While we acknowledge that the selection of these parameters influences the simulated cloud properties, their impact remains
limited when compared to the magnitude and temporal variability of the variables we analyzed.





**Figure 7.** Zonal means of cloud fraction for (a) LMDZ+COSP/AEOLUS, (b) AEOLUS (2020), (d) LMDZ+COSP/CALIPSO, (e) CALIPSO (2008), model-to-observations differences for (c) AEOLUS and (f) CALIPSO, model-to-model difference (g) between LMDZ+COSP AEOLUS and CALIPSO simulations, and observational difference (h) between AEOLUS and CALIPSO measurements.

## 4 Results


In this section, we assess whether the evaluation of cloud representation in the IPSL-CM6-LR model using the COSP-lidar simulator remains robust when based on observations from either CALIPSO or AEOLUS. First, we compared one year of observations from CALIPSO (2008) and AEOLUS (2020) to identify the main differences between the two measurement datasets, including those arising from their respective instrument configurations and measurement conditions. Second, we

analyzed the COSP-lidar simulations of the two configurations (COSP-lidar/CALIPSO and COSP-lidar/AEOLUS), in order to isolate differences due to the instrument designs. Finally, we evaluate the LMDZ model performance separately using




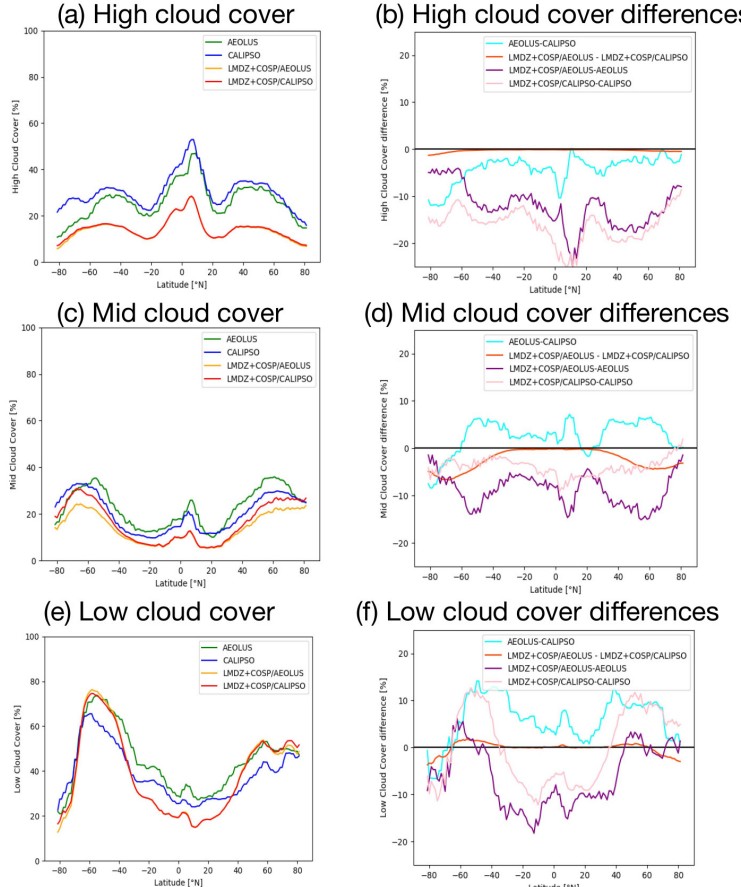

**Figure 8.** Zonal mean of yearly means of cloud covers and model-to-observations and model-to-model differences, at high (a,b) mid (c,d) and low (e,f) levels for AEOLUS and CALIPSO

CALIPSO and AEOLUS observations, and compare the resulting mode-to–observation discrepancies. This step accounts for both instrument-specific effects and the influence of interannual variability that have been previously discussed, aiming to determine whether cloud diagnostics derived from AEOLUS can be reliably used as CALIPSO for climate model evaluation.

## 4.1 AEOLUS vs CALIPSO observations

Figure 7 - middle column presents the zonal mean cloud fraction profiles derived from AEOLUS measurements in 2008 and CALIPSO observations in 2020, and their difference. Several key differences between AEOLUS and CALIPSO designs can lead to discrepancies in cloud detection that must be accounted for when comparing their retrievals. Firstly, AEOLUS uses a shorter wavelength than CALIPSO (Hunt et al., 2009) and CALIPSO is sensitive to polarization unlike AEOLUS - for which climatological depolarization ratios are used to compensate for better discrimination of cloud phase. Also, as we address in Section 3.3.a, their viewing geometries differ. Furthermore, CALIPSO and AEOLUS have limited co-locations due to their



| Global mean | COSP-lidar /AEOLUS | COSP-lidar /CALIPSO | OBS AEOLUS | OBS CALIPSO | COSP-lidar /AEOLUS - OBS AEOLUS | COSP-lidar /CALIPSO - OBS CALIPSO | COSP-lidar /AEOLUS - COSP-lidar /CALIPSO | OBS AEOLUS - OBS CALIPSO |
|---|---|---|---|---|---|---|---|---|
| High cloud cover | 13% | 14% | 26% | 30% | -13% | -16% | -1% | -4% |
| Mid cloud cover | 14% | 17% | 23% | 20% | -9% | -3% | -3% | +3% |
| Low cloud cover | 38% | 38% | 45% | 39% | -7% | -1% | - | +6% |

**Table 4.** Global mean of high/mid/low cloud covers from COSP simulations and observations for AEOLUS and CALIPSO, and their differences

| Global mean | COSP-lidar /AEOLUS | COSP-lidar /CALIPSO | OBS AEOLUS | OBS CALIPSO | COSP-lidar /AEOLUS - OBS AEOLUS | COSP-lidar /CALIPSO - OBS CALIPSO | COSP-lidar /AEOLUS - COSP-lidar /CALIPSO | OBS AEOLUS - OBS CALIPSO |
|---|---|---|---|---|---|---|---|---|
| High cloud cover | 13% | 14% | 25% | 27% | -12% | -13% | -1% | -2% |
| Mid cloud cover | 4.1% | 4.3% | 12% | 8.5% | -7.9% | -4.2% | -0.2% | +3.5% |
| Low cloud cover | 21% | 21% | 41% | 34% | -20% | -23% | - | +7% |

**Table 5.** Spatial mean in cumulus regions of high/mid/low cloud covers from COSP simulations and observations for AEOLUS and CALIPSO, and their differences

equatorial crossing times, which also introduces differences related to the diurnal cycle of clouds (Feofilov et al., 2024; Chepfer et al., 2019; Noel et al., 2018). To address this, a correction using diurnal variability observed by the CATS (Cloud–Aerosol Transport System) lidar onboard the ISS (International Space Station) could be used (Feofilov et al., 2024; Titus et al., 2025),

but we are not applying it in this study. The coarser horizontal resolution of AEOLUS can also play a role as it can lead to artificial increased cloud fraction due to the instrument's along orbit track resolution (Titus et al., 2025) especially in regions with sparse low level clouds that are seen as overcasted. AEOLUS averages the signal over larger volumes that can lead to a merge of smaller cloud signals, thus reducing the instrument's ability to detect small or thin cloud features.

The coarser horizontal resolution of AEOLUS further reduces sensitivity to small-scale clouds, particularly trade cumulus,

for which cloud fraction can be underestimated by up to 25% (Chepfer et al., 2013).



| Global mean | COSP-lidar /AEOLUS | COSP-lidar /CALIPSO | OBS AEOLUS | OBS CALIPSO | COSP-lidar /AEOLUS - OBS AEOLUS | COSP-lidar /CALIPSO - OBS CALIPSO | COSP-lidar /AEOLUS - COSP-lidar /CALIPSO | OBS AEOLUS - OBS CALIPSO |
|---|---|---|---|---|---|---|---|---|
| High cloud cover | 6.8% | 6.8% | 17% | 19% | -10.2% | -12.2% | - | -2% |
| Mid cloud cover | 3.8% | 4.1% | 7.7% | 6.5% | -3.9% | -2.4% | -0.3% | +1.2% |
| Low cloud cover | 44% | 44% | 58% | 51% | -14% | -7% | - | +7% |

**Table 6.** Spatial mean in stratocumulus regions of high/mid/low cloud covers from COSP simulations and observations for AEOLUS and CALIPSO, and their differences

| Global mean | COSP-lidar /AEOLUS | COSP-lidar /CALIPSO | OBS AEOLUS | OBS CALIPSO | COSP-lidar /AEOLUS - OBS AEOLUS | COSP-lidar /CALIPSO - OBS CALIPSO | COSP-lidar /AEOLUS - COSP-lidar /CALIPSO | OBS AEOLUS - OBS CALIPSO |
|---|---|---|---|---|---|---|---|---|
| High cloud cover | 17% | 17% | 34% | 35% | -17% | -18% | - | -1% |
| Mid cloud cover | 7.4% | 7.8% | 17% | 14% | -9.6% | -6.2% | -0.4% | +3% |
| Low cloud cover | 9.2% | 9.3% | 19% | 19% | -9.8% | -9.7% | -0.1% | - |

**Table 7.** Spatial mean in the Indian Ocean region of high/mid/low cloud covers from COSP simulations and observations for AEOLUS and CALIPSO, and their differences

A systematic larger cloud fraction (shown in red) by AEOLUS around 2000 meters altitude, that is bigger in the Southern Hemisphere, ranging from 8 to 20% (vs less than 12% in the Northern Hemisphere). This bias is consistent with expectations, as it likely results from the coarser spatial resolution of AEOLUS (3km along the orbit track) compared to the finer horizontal sampling of CALIPSO (330 meters) Two regions with smaller cloud fractions observed by AEOLUS (from 4 to 8%) are located between 40° and 60° in both hemispheres between 4000 and 10000 meters. As shown in the previous sensitivity tests (see Figure 5), this difference may result from the effects of viewing angle or multiple scattering. Elsewhere, the differences between the two lidars are less than 4% in absolute value and can be biased by interannual variability, as the observational years being compared are not the same.





| Global mean | COSP-lidar /AEOLUS | COSP-lidar /CALIPSO | OBS AEOLUS | OBS CALIPSO | COSP-lidar /AEOLUS - OBS AEOLUS | COSP-lidar /CALIPSO - OBS CALIPSO | COSP-lidar /AEOLUS - COSP-lidar /CALIPSO | OBS AEOLUS - OBS CALIPSO |
|---|---|---|---|---|---|---|---|---|
| High cloud cover | 32% | 32% | 52% | 56% | -20% | -24% | - | -4% |
| Mid cloud cover | 10% | 10% | 20% | 17% | -10% | -7% | - | +3% |
| Low cloud cover | 10% | 10% | 27% | 21% | -17% | -11% | - | +6% |

**Table 8.** Spatial mean in the Warm Pool region of high/mid/low cloud covers from COSP simulations and observations for AEOLUS and CALIPSO, and their differences

## 4.2 COSP-lidar/AEOLUS vs COSP-lidar/CALIPSO simulations

The differences between cloud fractions simulated using the AEOLUS and CALIPSO configurations are all below 10% in absolute value (Figure 7 (g)). Overall, the COSP-lidar/AEOLUS simulations produce lower cloud fractions (mostly around 4%) at all altitudes except in the low levels with respect to COSP-lidar/CALIPSO ones.

Figure 8 (g) is consistent with the sensitivity tests previously conducted (see Figure 5). It highlights the difference between the COSP-lidar/AEOLUS simulation configured with a wavelength of 355 nm, a multiple scattering efficiency factor n = 0.9,

and a viewing angle inclination of 35°, and the COSP-lidar/CALIPSO simulation using a wavelength of 532 nm, n = 0.7, and no inclination. In this case, the combined effects of a reduced multiple scattering coefficient and the absence of inclination result in a negative difference of approximately 4% in cloud fraction for mid-level clouds (between 4000 and 10000 meters) beyond 20° latitude and towards the poles. A positive difference in low-level cloud fraction, particularly over the Southern Hemisphere, is also observed, ranging from 4% to 6%. This difference arises from the specific characteristics of each instrument we accounted

for in the COSP-lidar algorithm (e.g. viewing angle and multiple scattering coefficient) as demonstrated by the sensitivity tests to various parameters presented in Section 3.3.

## 4.3 (MOD-OBS) AEOLUS vs (MOD-OBS) CALIPSO results and analysis

The model–observation differences for each instrument allow us to assess whether the model evaluation remains consistent regardless of the lidar used, provided that the COSP-lidar tool is appropriately configured and taking account of the actual

discrepancies between the two observational datasets. The previous comparisons demonstrated that the discrepancy between COSP-lidar/AEOLUS and COSP-lidar/CALIPSO (Figure 7 (g)) is consistent with the difference observed between the respective lidar measurements (Figure 7 (h)), particularly in the middle and upper troposphere, showing a small (around 4% and maximum 10%) underestimation of the cloud fraction for AEOLUS with respect to CALIPSO at every latitudes and altitudes.





Cesana et al. (2022) identified cloud phase and vertical distribution biases in CMIP6 models, which are consistent with the
cloud fraction biases we observe in our analysis. Similar spatial patterns emerge, suggesting that these biases persist across
datasets. The model evaluation presented in the figures 7 (c) and (f) share common features: a slight overestimation of high
level cloud fractions near 13000 meters at the equator (4-8%) by the model and a substantial underestimation of low level
cloud fractions below 2000 meters (more than 10%) across all latitudes but bigger in the Southern hemisphere (reaching up
to 20%) by the model. This underestimation is particularly high in cumulus and stratocumulus regions with a low cloud cover
bias higher in these areas (between 7 and 23%) than at the global scale (between 1 and 7%) (see tables 6, 7, 8). In the case
of CALIPSO, it should be noted that the model-observation difference can be affected by interannual variability but only by
a few percent (from 0.5 to 2%). It is crucial to highlight the negligible magnitude of the discrepancies linked to configuration
differences (presented in Section 4.2) in comparison with the amplitude of the model bias. Madeleine et al. (2020) reported
an overestimation of low-level cloud cover in the 30–60° latitude band of both hemispheres in IPSL-CM6A-LR, along with
a general underestimation elsewhere, based on comparisons with CALIPSO observations. Our results, illustrated in Figure
8 (g), corroborate these findings. In addition, we find that low and high-level cloud covers are globally underestimated in
IPSL-CM6A-LR, especially in the warm pool region by 11 to 24% (see tables ?? and ??).

Figure 8 (h) shows the differences between observational datasets from AEOLUS and CALIPSO. It is reassuring to observe
the recurrence of similar patterns as in Figure 8 (g) : a negative difference around 12000 meters at the equator, a negative
difference between 4000 and 1000 meters beyond 20° latitude toward the poles, and a positive difference near 2000 meters
altitude, consistent across all longitudes but more pronounced in the Southern Hemisphere. This suggests that the simulations
realistically reproduce the observed differences between the two instruments, which result from their respective measurement
configurations.

## 5  Conclusion and perspectives

This study highlights that despite AEOLUS not being designed for cloud measurements, its observations can serve like those
of CALIPSO as a valuable tool for evaluating cloud representation in General Circulation Models (GCMs). We have re-
assessed the significant underestimation of cloud fractions at low levels in the LMDZ model, which can be underestimated
with more than 20% bias (particularly in cumulus regions and in the southern hemisphere). High-altitude clouds are also un-
derestimated in specific regions such as the warm pool where the cloud cover negative bias can reach up to 20%.There is
no major difference (less than 4%) between the simulated cloud covers for AEOLUS (COSP-lidar/AEOLUS) and CALIPSO
(COSP-lidar/CALIPSO) configurations. We made sensitivity tests which explained that those small discrepancies are due to
viewing geometry, multiple scattering and sensivity (cloud detection threshold) differences between the two instruments. On
the observational side, comparisons between AEOLUS and CALIPSO measurements over a one-year period also reveal small
differences in cloud cover (less than 10%). These model-to-model and observation-to-observation differences are negligible
with respect to model biases. These findings underline the need for improved observational constraints and model parametriza-
tions for clouds and support the fact that model evaluations using AEOLUS are consistent to those using CALIPSO.



Looking ahead, a key challenge lies in merging long-term datasets from multiple spaceborne lidars, incorporating successively CALIPSO (2006-2023), AEOLUS (2018-2023) and ATLID (since 2024). This requires harmonized processing strategies for the measurements datasets and adapted configurations of the COSP-lidar tool to ensure continuity across the instruments for reliable multi-decades model-to-observation comparisons. As we successfully developed the AEOLUS module in COSP - used here as a reference for EarthCARE/ATLID due to their shared characteristics and the temporal overlap in measurements with CALIPSO - future work will focus on refining again the COSP-lidar algorithm to perform similar simulations for Earth-CARE/ATLID. We aim to establish a comprehensive multi-lidar comparison with CMIP6 model outputs of cloud observations from CALIPSO to EarthCARE/ATLID. Furthermore, since ATLID is specifically designed for cloud detection and offers a fine vertical resolution, similar cloud-related biases in the model are expected to be observed with potentially greater amplitudes.

Additionally, AEOLUS wind measurements (not used in the current study) offer a unique opportunity in future work to assess the vertical and global performance of modelled winds, and to explore cloud-wind interactions in GCMs through the COSP/AEOLUS framework.

*Code availability.* The updated version of the COSP-lidar algorithm, including the developments related to AEOLUS will be made publicly available on the official COSP GitHub repository upon publication of the article.

*Author contributions.* MLR and HC conceived and designed the study. Data collection was performed by MLR and ZT. All the authors interpreted the results. The manuscript was written by MLR and HC with input from all co-authors.

*Competing interests.* The authors declare that they have no conflict of interest.

*Acknowledgements.* The authors thank NASA and CNES for providing CALIPSO satellite data and ESA for providing AEOLUS satellite data. We acknowledge the CNRS for the funding of MLR.



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



|  | Variable name in COSP | Description |
|---|---|---|
| **Optical inputs** | beta_mol_aeolus | Lidar molecular backscatter coefficient |
|  | betatot_aeolus | Lidar backscatter coefficient |
|  | tau_mol_aeolus | Lidar molecular optical depth |
|  | tautot_aeolus | Lidar molecular optical depth |
| **Logical** | Laeolus_column | ON to use the lidar column routine for AEOLUS |
|  | Laeolus_subcolumn | ON to use lidar subcolumn routine for AEOLUS |
|  | ok_lidar_cfad_aeolus | ON to compute lidar CFAD diagnostics |
| **Outputs** | aeolus_beta_tot | Total backscattered signal |
|  | aeolus_beta_mol | Molecular backscatter |
|  | aeolus_cfad_sr | CFAD of scattering ratio |
|  | aeolus_lidarcld | 3D cloud fraction |
|  | aeolus_cldlayer | low, mid, high and total cloud cover |
|  | aeolus_srbval | SR bins in cfad_sr |

**Table A1.** Description of the new variables included in the Aeolus version of COSP-LIDAR

## Appendix A: Additional technical details on the COSP-lidar/AEOLUS implementations

An "AEOLUS interface" has been implemented to define the AEOLUS-specific Fortran data type and to provide a dedicated, currently empty, initialization routine. The following table A1 provides the description of the new AEOLUS-specific variables added to the COSP algorithm.

The table A2 gives the list of the variables that are mandatory to run the COSP-lidar simulations in our developpement mode.



| Variable name in COSP | Variable name in IPSL-CM6 | Variable name in CMIP6 database | Dimension | Description | Frequencies available |
|---|---|---|---|---|---|
| uLOSaeolus | uLOSaeolus | ua | 3D | northward wind | CFday |
| vLOSaeolus | vLOSaeolus | va | 3D | eastward wind | CFday |
| psfc | psol | ps | 2D | surface pressure | CFday |
| pfull | pres | pfull | 3D | air pressure (full levels) | CF3hr, CFday |
| phalf | paprs | phalf | 3D | air pressure (mid-levels) | CF3hr, CFday |
| height | zfull | zfull | 3D | altitude of full pressure levels | CF3hr |
| height_half | zhalf | zhalf | 3D | altitude of half pressure levels | CF3hr |
| T_abs | temp | ta | 3D | air temperature | CF3hr, CFday |
| qv | ovap | hus | 3D | specific humidity | CF3hr, CFday |
| rh | rhum | hur | 3D | relative humidity | CFday |
| tca | rneb | cl | | cloud fraction | CFday |
| cca | rnebcon | clc | | convective cloud fraction | CF3hr |
| mr_lsice | icc3dstra | clis | | mass fraction of stratiform cloud ice | CF3hr |
| mr_ccliq | lcc3dcon | clwc | | convective cloud liquid fraction | CF3hr |
| mr_ccice | icc3dcon | clic | | mass fraction of convective cloud ice | CF3hr |
| fl_lsrain | pr_lsc_i | prlsprof | 3D | large scale precip liquid | CF3hr |
| fl_lssnow | pr_lsc_i | prlsns | 3D | large scale precip ice | CF3hr |
| fl_ccrain | pr_con_i | prcprof | 3D | convective precip liquid | CF3hr |
| fl_ccsnow | pr_con_i | prsnc | 3D | convective precip ice | CF3hr |
| skt | tsol | ts | 2D | surface temperature | CF3hr |
| orography | phis | orog | 2D | surface geopotential height | fx |
| landmask | contfracATMO | sftlf | 2D | % land surface | fx |
| Reff | ref_liq, ref_ice | reffclis, reffclws | 3D | | CF3hr |

**Table A2.** Description of the model variables that are mandatory as inputs of COSP simulations