# Peer review of "GCM clouds and actual clouds as seen from different space lidars : towards a long-term assessment of cloud representation in GCMs using lidar simulators"

_EGUsphere, 2025_

## Author Comment (AC1)

**Reply to Anonymous referee #1**

All co-authors thank the reviewer for the comments and corrections which helped improve the manuscript.

*Line 20: Including references to further works on cloud radiative feedbacks, particularly from older CMIP generations (CMIP3 Bony & Dufresne (2005), CMIP5 Vial et al. (2013)) would help contextualize how big the issue is.*

*Bony, S., & Dufresne, J.-L. (2005). Marine boundary layer clouds at the heart of tropical cloud feedback uncertainties in climate models. Geophysical Research Letters, 32(20), L20806.*

*Vial, J., Dufresne, J. L., & Bony, S. (2013). On the interpretation of inter-model spread in cmip5 climate sensitivity estimates. Climate Dynamics, 41(11–12), 3339–3362.*

**Those references have been added to the publication list and in the main text of the manuscript to enhance the literature background of the paper.**

*Line 31: This paragraph is confused by the introduction of an objective in the paper in the middle of introducing COSP. I would suggest rounding out the paragraph on COSP and moving everything following "Extending the COSP-lidar..." to the beginning of the next paragraph.*

*Line 36: Introducing the overarching goal that this study contributes towards is important, however this message is interspersed throughout the introduction, causing a bit of confusion. The paragraph beginning on line 36 is around the overarching goal this study contributes to. The following paragraph beginning line 44 is around the LDMZ model specific to this study. Then the paragraph beginning line 51 begins with the objective of this study before again reiterating the long term goal. These need to be separate paragraphs for clarity, one being what is happening in this paper, the other being the long term goal this work contributes towards.*

**We have separated the overall goals of this kind of study and the goals of the paper properly in the introduction to clear up the individuals objectives.**

*Line 82: The description of CALIPSO is excellent. However, the paragraphs beginning lines 82/89 are describing data selection for the experimental procedure which is detailed further in section 2.3. I would prefer this information to be in section 2.3 as the reader doesn't yet know why we are talking about these individual years yet.*

**We moved this paragraph to the section 2.3.**

*Section 2.3: Multiple specific years and dates ranges are used in this section. A lot of effort has been put into making sure these two individual years are comparable, but no readily apparent justification is given as to why were only looking at these individual years. Please see the following specific comments.*

*Line 114: Why these years specifically? In section 2.1 it is stated that 2008 data from CALIPSO is used to for consistency with simulations, **but why not simulate 2020,** and compare to CALIPSO 2020 and AEOLUS 2020. I assume due to the lack of AMIP forcing data.*

*Further to the previous point, why then cant a comparison between the 2008-2018 CALIPSO climatology and 2018-2023 AEOLUS climatology be made, thus avoiding the need to compare individual years and worry too much about inter-annual variability?*

*Is only 2020 produced by Titus et al. (2025)? This would need to be stated. If so comparisons of AEOLUS 2020 to the CALIPSO 2008-2018  or CALIPSO 2020 could be made.*

*Line 117: So the effect of interannual variability is only assessed using the CALIPSO product and assumed to be the same for AEOLUS. Based on the results presented further in the manuscript these products are highly comparable and would thus be fine, but  would be worth mentioning at this stage. A supplementary figure detailing where 2020 fits within the inter-annual variability of 2008-2018 would be good.*

*It would also be worth mentioning that 2008 and 2020 are both La Nina years in terms of comparing clouds in the pacific.*

**We added information about the lack of CMIP6-AMIP data and the context of La Nina in section 2.3.**
**We have also mentioned in the section 2.3 of the manuscript that the interannual variability of clouds estimated from CALIPSO-GOCCP observations over 2008–2018 is expected to be of the same order of magnitude as AEOLUS. Nevertheless, we also specify in the paper that a more complete CALIPSO–AEOLUS comparison (accounting for differences in the diurnal cycle and horizontal resolution) is available in the literature and is not the main focus of our study.**

*Line 194: The multiple scattering coefficient n is what is different between Figure 5 (b) and (d), not the inclination. Do you mean (a) and (b), or, (c) and  (d)?*

**This is an error that is now corrected in the new version of the paper.**

*Line 331: the sentence beginning: "We have reassessed the significant underestimation of cloud fractions at low levels in the LMDZ model" should be moved to later in the conclusion, as the focus should be on the AEOLUS-CALIPSO comparison.*

**This sentence has been moved later in the conclusion to better separate the various points mentioned in it.**

*Discussion point: Only one GCM was evaluated upon here. How would results be different if a different GCM was evaluated upon?*

**If a different GCM is evaluated, we expect that the results may be in accordance with the actual performances of the model. The model bias patterns should be consistently identified in both CALIPSO and AEOLUS comparisons, since the use of COSP is mostly model-independent. Moreover, several evaluations of other GCMs against CALIPSO observations have already been conducted using COSP in previous studies (Cesana et al. 2024; Konsta et al. 2022; Morrison et al. 2018).**

Cesana, G.V., Ackerman, A.S., Fridlind, A.M. *et al.* Observational constraint on a feedback from supercooled clouds reduces projected warming uncertainty. *Commun Earth Environ* **5**, 181 (2024). https://doi.org/10.1038/s43247-024-01339-1

Konsta, D., Dufresne, J.-L., Chepfer, H., Vial, J., Koshiro, T., Kawai, H., et al. (2022). Low-level marine tropical clouds in six CMIP6 models are too few, too bright but also too compact and too homogeneous. *Geophysical Research Letters*, 49, e2021GL097593. https://doi.org/10.1029/2021GL097593

Morrison, A. L., Kay, J. E., Frey, W. R., Chepfer, H., & Guzman, R. (2019). Cloud response to Arctic Sea ice loss and implications for future feedback in the CESM1 climate model. *Journal of Geophysical Research: Atmospheres*, 124, 1003–1020. https://doi.org/10.1029/2018JD029142

*Technical Corrections:*

*Title: should the last word of the title be the plural of simulator: ....using lidar simulators*

*Line 6: It would be good to specify that ALADIN is the instrument aboard AEOLUS*

*Line 10: Single "*

*Line 68: "layer height" is this a common variable? I would expect a mention of cloud coverage here as well*

*Line 181: The inclinaison used instead of inclination.*

*Line 185: The following section would then be 3.4?*

*Line 266: Shouldn't this be CALIPSO observations in 2008 and AEOLUS observations in 2020.*

*Line 293: Shouldn't this be Figure 7 (g)?*

*Line 321 and furthermore: Figure 8 doesn't have a panel (g) or (h).*

*Line 323: Table numbers appearing as ??.*

**We thank you for pointing out these mistakes. The corrections have been made in the paper.**

---

## Author Comment (AC2)

**Reply to Anonymous referee #2**

All co-authors thank the reviewer for the comments and corrections which helped improve the manuscript.

*Definition of cloud fraction in 3 layers. My understanding is that GOCCP low-, mid-, and high-level cloud fraction is based on pressure layers: low-level (P > 680 hPa), middle-level (440 < P < 680 hPa) and highest-level (P < 440 hPa). However, in this study height is used instead of pressure (Lines 120-122, Table 1). It is not clear if a different type of processing has been done for this study. This needs clarification.*

**In the new version of the manuscript, we reprocessed AEOLUS low-, mid-, and high-level cloud covers based on pressure layers, in order to stay consistent with CALIPSO-GOCCP low-, mid-, and high-level cloud covers. The pressure layer thresholds are still P < 440 hPa for high-level clouds, 440 < P < 680 hPa for mid-level clouds, and P > 680 hPa for low-level clouds.**

*Section 2.3. I'm confused by this section. CALIPSO and AEOLUS overlap, so why not compare cloud retrievals during the overlapping period? That would avoid the impact of internal variability.*

**This section has been elaborated to clarify this specific point : the AEOLUS product we use is only made for 2020 and the CALIPSO-GOCCP product is affected by low laser shots after 2018 leading us to only take into account the 2008-2018 decade for inter-annual variability of clouds.**

*Use of daily-averaged input data for COSP calculations. There is no discussion on the impact of this choice. A better motivation for this choice is needed. It would be interesting to see differences between daily averages calculated from 3-hourly inputs and daily-averaged inputs. Are these differences small enough with respect to other sources of uncertainty considered?*

**Other studies have been conducted with LMDZ (Chepfer et al. 2008) in the literature and have shown that the impact of using averages of hourly or three-hourly outputs, compared to daily means, is negligible in the assessment of model biases. Moreover, the diurnal variability expected is much smaller than the model bias itself (for a comparison between AEOLUS and CALIPSO accounting for the diurnal cycle, see Titus et al., 2025).**

*I may be misinterpreting the results, but Figure 7h doesn't seem to match with the results shown in Figure 8. Figure 7h shows less AEOLUS cloud fraction nearly everywhere above between 4000m and 8000m altitude, but Figure 8d shows an excess of AEOLUS mid-level cloud with respect to CALIPSO.*

**This was an error in the altitudes that have been taken into account during the processing of the cloud covers. We corrected it and updated the figure 8. The results of the mid cloud covers observed by AEOLUS are now in accordance with the cloud fractions shown in the figure 7.**

*L45: that is involved -> that participates*

*L70: inclinaison. Typo that appears many times in the manuscript. I'd recommend replacing all instances of 'inclination' with 'off-nadir pointing angle' to avoid confusion with the inclination of the satellite orbit.*

*L122: global median of high cloud cover -> median of global-mean high-cloud cover. There are other instances in the text, please correct.*

*L192: cloud detection threshold s. Capital S is used before in reference to the detection threshold. Please use consistent notation.*

*Table 2. Please use Greek letters for 'eta' and the wavelength 'l'.*

**We thank you for pointing out these technical issues. We have taken them into account in the paper.**